# Propagation of *Calendula maritima* Guss. (Asteraceae) through Biotechnological Techniques for Possible Usage in Phytotherapy

**Caterina Catalano** [ID]**, Loredana Abbate** [ID]**, Francesco Carimi** [ID]**, Angela Carra \*** [ID]**, Alessandro Silvestre Gristina** [ID]**, Antonio Motisi, Salvatore Pasta** [ID] **and Giuseppe Garfi** [ID]

Institute of Biosciences and BioResources (IBBR), National Research Council (CNR), Research Division of Palermo, Corso Calatafimi 414, 90129 Palermo, Italy

* Correspondence: angela.carra@ibbr.cnr.it

**Abstract:** The genus *Calendula* (Asteraceae) includes several species that are renowned for their therapeutic properties and/or use as ingredients in the preparation of cosmetics. The rare and critically endangered sea marigold, *Calendula maritima* Guss., an endemic species from Western Sicily, has also been recognized as a potential "farm plant species" for several important compounds used in cosmetics. However, the few remnant populations of this species are currently threatened with extinction because of several factors, such as hybridization with the congeneric species *Calendula suffruticosa* subsp. *fulgida* (Raf.) Guadagno and anthropogenic disturbance of its habitat. Therefore, in order to preserve the genetic integrity from pure genetic lineages, seed-based propagation and seed storage are not recommended for either conservation or massive production purposes. In this paper, we describe a protocol adopted for mass propagation of *C. maritima* from selected genotypes. Nodal segments collected from selected plants growing in the field were used as starting explants and were cultured for micropropagation on MS medium with and without phloroglucinol. New shoots produced were cultured for rooting under several conditions with the aim of finding the best procedure favoring root induction. The best rooting performance was obtained with zeolite and rooted plants were successfully acclimatized outdoors. The technique described allowed the multiplication of genotypes of interest as well as to overcome the problems of hybridization of this species, hence contributing concretely to the conservation of the sea marigold.

**Keywords:** Asteraceae; in vitro propagation; secondary metabolites; massive plant production; genetic resources' conservation; Mediterranean vascular flora; threatened plants; root induction



## 1. Introduction

*Calendula maritima* Guss., the sea marigold, is a rare and endemic species belonging to the *Asteraceae* family, only growing along the coast between Marsala and Trapani and some satellite islets of Western Sicily. According to IUCN criteria, it is considered "CR" (Critically Endangered) because of several factors [1]. The main threats are related to the conservation of its natural habitat, which is extremely degraded and fragmented [2]. Additionally, many populations are rather unsteady, very small, and severely threatened by competition owing to invasive alien plants like *Carpobrotus edulis* (L.) N.E.Br. and by ongoing hybridization with the congener *Calendula suffruticosa* subsp. *fulgida* (Raf.) Guadagno [3]. Moreover, many populations grow in areas with a strong human impact related to seasonal recreational activities.

The genus *Calendula* includes several species that have been reported for their therapeutic properties and/or use as ingredients for food and purposes of cosmetics' preparation. Within the genus, *C. officinalis* L. is the most popular and studied for phytochemistry and ethnopharmacological aspects [4], while until now, little information about *C. maritima* is available. *C. maritima* has been recognized as a potential "farm plant species" owing to some important compounds that are of interest for the cosmetic industry; moreover,

some investigations recently pointed out that leaf extracts show high antioxidant activity [5], whereas the essential oil extracted from aerial parts is proven to have insecticidal properties [6].

Considering the potential use of *C. maritima* for the industrial production of bioactive compounds of phytotherapeutical interest, it is important to establish an effective procedure for massive production purposes without affecting the shrinking wild populations. For this reason, vegetative propagation using cuttings is not an effective procedure. Moreover, seed-based propagation and seed storage are not recommended because of the hybridization problems cited above [7]. In situ conservation and management is, nevertheless, quite difficult. *C. maritima* is a nitrophilous plant growing in areas subject to intense disturbance, habitat disruption, and urbanization; prone to coastal erosion; and degraded by human pressure owing to seasonal disturbances connected with seaside tourism [8]. In such critical situations, in vitro propagation techniques offer a valid biotechnological tool for plant conservation and production [9,10]. Micropropagation plays a key role in the production and ex situ conservation of rare, endemic, and endangered plants [11,12]. Additionally, it can be of great commercial and economical importance [13] because it allows to obtain, in a short time, large amounts of the selected genotypes starting from a minimum stock of living material gathered in the field, without involving significant changes to the conservation status of wild populations [14]. A regeneration protocol by direct organogenesis from leaves of *C. maritima* has been already provided by Carra et al. [7]. However, as the regeneration procedure usually passes through a de-differentiation and re-differentiation phase, it is time consuming. Moreover, as a callus phase is usually present, it is preferable to verify the genetic fidelity of the regenerants. In order to overcome such an inconvenience, in the present paper, we report a new in vitro procedure aimed at producing large numbers of sea marigolds bypassing the callus phase. In vitro multiplication was performed starting from nodal explants in the best growing conditions. Moreover, as the success of plant tissue culture depends on its potential to transfer plants in the field for large-scale production with a high survival rate, the acclimatization step is the crucial point of the entire protocol [15]. For this reason, a detailed description of rooting and acclimatization procedure is also reported, with a special focus on the substrates used for the rooting step. The new plants thus obtained can be used both for production purposes as well as for reintroductions into the wild in the framework of conservation projects.

## 2. Materials and Methods

### 2.1. Plant Material and Culture Condition

Preliminary field surveys based on flower and leaf morphology were carried out to avoid sampling hybrids between *C. maritima* and *C. fulgida*. Actively growing shoots were collected in the springtime from mature plants of *C. maritima* belonging to populations located in the locality Ronciglio south of Trapany city (Lat. 38°00′33″ N, Long. 12°30′36″ E) and satellite islets of Maraone (37°59′23″ N, 12°24′51″ E) and Colombaia (38°00′42″ N, 12°29′32″ E), which previous investigations indicated as being the purest genetic lineages (G. Garfì, *pers. comm.*), and used as explant source. For axenic culture establishment, leaves were removed from shoots, and the nodal explants (3–4 cm long) were rinsed under running tap water for 15 min and subsequently soaked for 15 min in distilled water with 3 drops/100 mL of Tween−20 with gentle shaking. Then, the explants were surface sterilized with ethanol 70% for 1 min and subsequently in 2% (*w/v*) sodium hypochlorite solution for 15 min, followed by three 5 min rinses in sterile distilled water under laminar flow (Figure 1A). Explants were incubated on Murashige and Skoog [16] solidified medium (MS) (8 gL$^{-1}$ Phytoagar, Duchefa) with 88 mM sucrose as a carbon source. The pH of the media was adjusted to 5.7 ± 0.1 with 0.5 M potassium hydroxide (KOH) before autoclaving. Plant Preservative Mixture ® (PPM, Plant Cell Technology, Washington, DC, USA) was supplemented to the medium at 2% after autoclaving in order to prevent contaminations. For axenic culture establishment, shoot multiplication, plant development, and rooting, we used Magenta™ glass vessels for plant tissue culture (Sigma, St. Louis, MO, USA, V-0633).

Explants were incubated in a climate chamber at $25 \pm 1$ °C under a 16 h day length and a photosynthetic photon flux of 50 µmol m$^{-2}$ s$^{-1}$ provided by Osram cool-white 18 W fluorescent lamps.

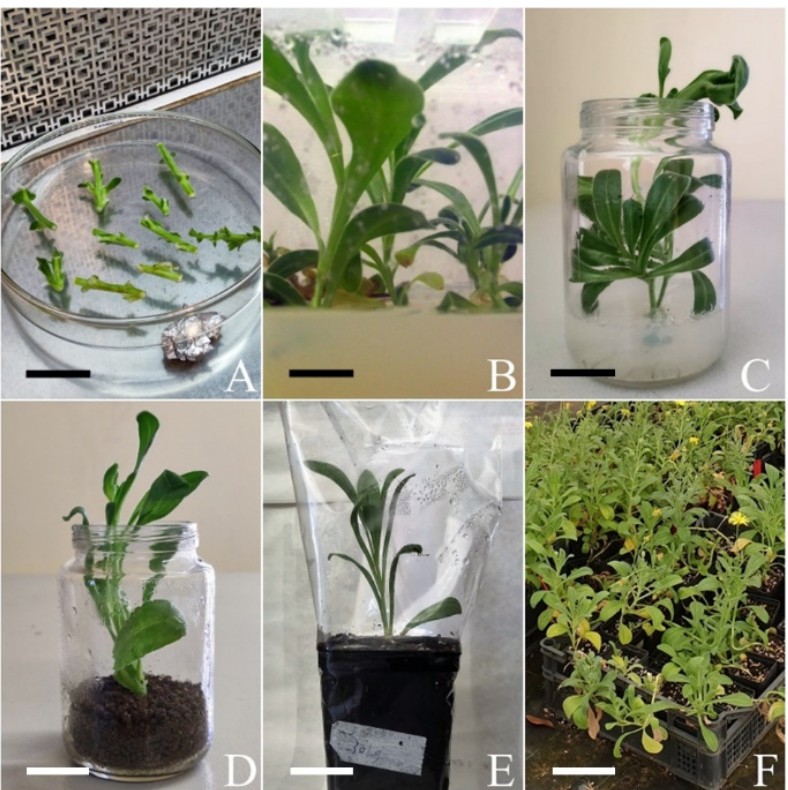

**Figure 1.** In vitro procedure for *C. maritima* plant regeneration from nodal explants. (**A**) Mono and binodal explants used for in vitro introduction (bar: 1 cm). (**B**) In vitro shoot regeneration obtained on MS medium after 30 days of culture (bar: 1 cm). (**C**) In vitro rooting obtained on MS medium after 30 days of culture (bar: 2.5 cm). (**D**) In vitro rooting obtained on Zeolite after 30 days of culture (bar: 2.5 cm). (**E**) Acclimatization phase (bar: 3.5 cm). (**F**) In vitro regenerated plantlets of *C. maritima* acclimatized in outdoor conditions after 30 days from transfer in outdoor conditions (bar: 7 cm).

*2.2. Shoot Multiplication*

In order to obtain enough plant material for the rooting step, axillary shoots (2–3 cm in length) were excised from two-week-old stem tissue and subcultured under two different culture conditions. Explants were plated in MS medium supplemented with 2% PPM and 3 µM phloroglucinol (PG) (1,3,5-trihydroxybenzene, Sigma, St. Louis, MO, USA, 79330). To evaluate the effect of PG on shoot formation, explants were also plated in MS medium supplemented only with 2% PPM. Explants were then subcultured at 30-day intervals and maintained in a climate chamber under the same culture conditions described above. The effect of different culture conditions on shoot proliferation was observed after 30 days of incubation, recording the percentage of responsive explants, the number of new shoots for each initial explant, the shoot length, and the number of leaves. The first three parameters were used to calculate a new parameter in order to evaluate the performance of explants cultivated under two different culture conditions [17]. This parameter, the "shoot elongation efficiency" (SEE), represents the total length (cm) of vegetation produced in vitro on average from one explant and is calculated by the formula SEE = (Percentage of explants producing shoots × Number of shoots per explant × Mean shoot length)/100.

### 2.3. Rooting and Plant Acclimatization

Green shoots (2–3 cm long) (Figure 1B) were collected at the end of 30 days of micro-propagation and used for in vitro rooting. To select the best rooting conditions, two sets of experiments were performed.

**First set**: shoots were cultured on solidified MS medium (8 g L$^{-1}$ Phytoagar, Duchefa) with 88 mM sucrose, in constant presence of 3-indoleacetic acid (IAA, Sigma, St. Louis, MO, USA, I-2886) at 5 and 10 μM. The pH of the medium was 5.7 ± 0.1. Medium was also supplemented with 2% PPM and 3 μM PG (Figure 1C).

Based on the preliminary results obtained from shoots rooted as described above, a second set of rooting tests was performed.

**Second set**: shoots were cultured on mineral zeolite (a natural aluminium silicate with excellent ionic exchange properties and a high power of absorption), according to Rosell et al. [18]. To each Magenta™ glass vessel, 57 g of autoclaved zeolite and 10 mL liquid MS medium supplemented with 5 μM IAA, 2% PPM, and 3 μM PG were added (Figure 1D). One explant per magenta was cultured.

Shoots incubated directly on MS medium or zeolite deprived of IAA and PG were used as a control. The effect of treatments was evaluated by recording the percentage of rooted shoots, the average number of roots per explant, and the average length of the roots. Shoots that formed roots were collected 30 days after rooting treatments and washed with tap water in order to remove the medium before being transplanted individually into plastic pots of 70 × 70 × 100 mm containing sterile soil. The potted plants, covered with transparent polyethylene bags to maintain temperature and high humidity (Figure 1E), were placed in a climate chamber at 25 ± 1 °C under a 16 h day length and a photosynthetic photon flux of 50 μmol m$^{-2}$ s$^{-1}$ provided by Osram cool-white 18 W fluorescent lamps. After 20–30 days, plantlets were exposed to gradual reduction of humidity and, after 40 days, plants were transferred outdoors under natural daylight conditions, but sheltered by a shadowing net for the final acclimatization (Figure 1F). The survival rates were recorded after 2 months.

### 2.4. Data Analysis

Treatments were carried out with 50 explants and repeated twice. Each treatment comprised 50 explants cultured singularly in a magenta vessel. The effect of different media on propagation and rooting was observed after 30 days of incubation. The experimental design was completely randomized and the effects of culture conditions were tested by analysis of variance (ANOVA) ($p \leq 0.05$); differences among means were tested by Tukey's test ($p \leq 0.05$). When effects of culture conditions were expressed as percentages, data were arcsin-square root transformed prior to analysis.

## 3. Results

### 3.1. Shoot Multiplication

In the first step, young nodal explants collected in the springtime were introduced in vitro from mature material and cultivated on MS medium without growth regulators, until enough stock material was available. The percentage of explant contamination was about 45%, in line with the period of collection (springtime). Shoot formation started within 7 days after culture initiation (Figure 1B) and shoot production was obtained both with and without PG, without significant differences according to cultural conditions (Table 1). The percentage of responsive explants ranged from 60.9% (MS + 3 μM PG) to 67.3% (MS). The average number of new shoots per responsive explant varied from 1.5 (MS + 3 μM PG) to 2.1 (MS), with no significant differences among treatments. The best result in terms of mean shoot length (3.4 cm) was obtained with explants cultured in presence of 3 μM PG. The SEE indicates that MS medium without PG was more effective in shoot multiplication even if the difference between treatments was not significant.

**Table 1.** Effect of different culture conditions on axillary bud formation in *C. maritima* explants.

| Treatment | Explants Producing Shoots (%) | No. of Shoots per Explant | Mean Shoot Length (cm) | Mean Leaves Number | SEE |
|---|---|---|---|---|---|
| MS | 67.3 ± 11.5 [a] | 1.8 ± 0.3 [a] | 3.1 ± 0.3 [a] | 3.9 ± 0.3 [a] | 3.8 [a] |
| MS + 3 μM PG | 60.9 ± 3.7 [a] | 1.5 ± 0.2 [a] | 3.4 ± 0.3 [a] | 3.7 ± 0.3 [a] | 3.1 [a] |

The effect of treatments on axillary bud formation is expressed as percentages of primary stem segment producing shoots, the number of shoots per explant, the average length of the newly regenerated shoots, and the average number of leaves produced. Data were collected after 30 days from the beginning of the experiment and each treatment comprised 50 explants. Means ± SE. Within columns, different letters indicate significant differences (LSD test, $p \leq 0.05$). MS: shoots incubated in MS medium hormone free. MS + 3 μM PG: shoots incubated in MS medium supplemented with 3 μM PG. SEE (shoot elongation efficiency) represents the total length (cm) of vegetation produced in vitro on average from one explant.

### 3.2. Rooting and Acclimatization

Healthy growing shoots were cultured in Magenta vessels under the culture conditions reported in Table 2. Roots emerged from the cut surface of cuttings about 15 days after treatments. The effect of each treatment was determined by recording the percentage of rooted shoots, the number of roots per explant, and the mean length of the roots (Table 2). Explants produced roots under all of the culture conditions tested, with the results varying according to the treatment (Table 2). Percentages ranged from 33.3% to 87% (in 10 μM IAA and zeolite + 5 μM IAA, respectively). In terms of number of roots per explant, there was no significant difference between treatments and, for all of the rooting conditions, the value was about 1.5 root per explant. The best results for root elongation (3.8 cm) were achieved in the presence of 5 μM IAA. Significantly different roots are produced depending on the medium used; i.e., roots induced in zeolite grew up in several waves in the radial direction and they were markedly shorter and thicker than those obtained with MS medium as support (Figure 2A–D). The first step of acclimatization was performed in a growth chamber. Well-developed plants were gradually exposed to a lower relative humidity and a higher light intensity. About 40 days after ex-flasking, plantlets reached about 15 cm in height (Figure 1F). The highest percentage of acclimatized plants (87.7%) was obtained with shoots rooted in zeolite + 5 μM IAA. The plantlet survival rate during acclimatization and transfer to greenhouse conditions was about 75%.

**Table 2.** Effect of different rooting treatments on root formation in *C. maritima* explants.

| Treatment | Rooted Plants (%) | No. of Root per Explant | Mean Root Length (cm) | Survival Rate of Rooted Plants (%) |
|---|---|---|---|---|
| Zeolite + 5 μM IAA | 87.1 ± 4.5 [a] | 1.4 ± 0.03 [a] | 2.6 ± 0.4 [b] | 87.7 ± 3.1 [a] |
| MS + 5 μM IAA | 52.5 ± 13.1 [b] | 1.5 ± 0.2 [a] | 3.8 ± 1.3 [a] | 28.7 ± 6.1 [c] |
| MS + 10 μM IAA | 33.3 ± 6.7 [c] | 1.5 ± 0.5 [a] | 3.1 ± 0.1 [b] | 17.3 ± 7.0 [c] |
| Hormone Free | 49.8 ± 6.1 [c] | 1.6 ± 0.3 [a] | 2.5 ± 0.5 [b] | 59.1 ± 8.6 [b] |

The rooting response is expressed as a percentage of rooted shoots, the number of roots per explant, and the mean root length. The percentage of surviving plants 40 days after transfer in greenhouse conditions is also reported. Data were collected after 40 days from the beginning of the experiment and each treatment comprised 50 explants. Means ± SE. Within columns, different letters indicate significant differences (LSD test, $p \leq 0.05$). Zeolite + 5 μM IAA: shoots incubated in zeolite supplemented with 5 μM IAA. MS + 5 μM IAA: shoots incubated in MS medium supplemented with 5 μM IAA. MS + 10 μM IAA: shoots incubated in MS medium supplemented with 10 μM IAA. Hormone free: shoots incubated in MS medium deprived of growth regulators.

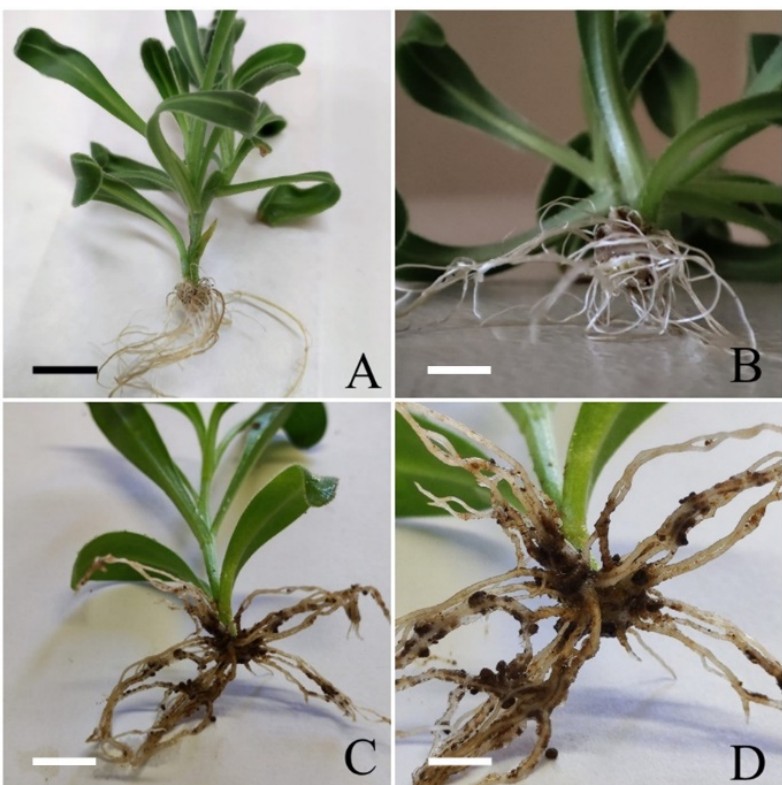

**Figure 2.** Root formation under different culture conditions after 30 days of culture. (**A**) Rooted plant obtained on MS medium supplemented with 5 μM IAA (bar: 2 cm). (**B**) Particulars of root apparatus obtained on MS medium (bar: 0.5 cm) (**C**) Rooted plant obtained on Zeolite supplemented with 5 μM IAA (bar: 2 cm). (**D**) Particular of root apparatus obtained on Zeolite + 5 μM IAA (bar: 0.5 cm). All pictures were taken after 30 days of culture.

## 4. Discussion

Many biologically active chemical compounds utilized in the pharmaceutical industry are traditionally derived from plants. Secondary metabolites are the most valuable phytochemicals, but the majority of them have complex structures that make their industrial synthesis extremely challenging [19,20]. In the aim to produce and accumulate large quantities of chemical compounds of interest to the pharmaceutical and medical industry, plant in vitro propagation offers several advantages [21,22]. The in vitro production reduces the environmental interference due to controlled growth conditions, allows the possibility to better manage the production of secondary metabolites through the development of step-by-step protocols, enables a season-independent production of molecules of interest, and allows the production under sterile conditions with very few risks [23]. Additionally, as reported for several species [24], biotechnology may be effective in increasing, in terms of both quality and production, the synthesis of secondary metabolites in medicinal plants.

The *Asteraceae* family, which includes more than 1600 genera and 2500 species worldwide, is one of the largest families of flowering plants [25]. Since ancient times, its representatives have been employed in food and medicinal preparations, and most of them, although having a broad range of differences, have a similar chemical composition [26]. For instance, all species show strong antibacterial, antioxidant, and anti-inflammatory activities, as well as diuretic and wound-healing qualities [25]. These properties are well renowned for *C. officinalis* but are very likely also present in the sea marigold [27]. Here, we report a successful in vitro procedure that allows the fast production of large numbers of *C. maritima* individuals independently from the species' life cycle. Propagation of *C. maritima* was obtained with no major hurdles after the in vitro establishment was achieved. As has been shown for many species [28–30], explants collected from mature plants growing in the field

may be difficult to introduce in vitro because the beginning of axenic cultures is limited by the insurgence of bacterial or fungal contamination. For this reason, it is often desirable to utilize young, actively developing shoots that were picked in the spring to prevent material loss due to culture contamination [17]. The use of young explants and an extensive surface sterilization procedure may not be enough to ensure an effective establishment of axenic culture because of the presence of endophytes that can survive surface sterilization [31]. PPM has been described as an effective biocide/fungicide for plant tissue culture and has been successfully used on several crops to prevent culture contamination [32,33].

Few protocols about *Calendula* micropropagation are available and they are mostly focused on *C. officinalis.* These reports suggest that the best results, in terms of multiple axillary shoot induction, were obtained when cytokinins, alone or in combination with low concentrations of auxins, were used [34,35]. Actually, the selection of plant growth regulators influences the explant's responsiveness and the morphogenic reaction. The composition of the culture medium's growth regulators can affect how frequently plants undergo morphological and physiological changes [36]. Auxins and cytokinins, in particular, are prominent sources of stress that have been connected to recalcitrance, habituation, somaclonal variation, and hyperhydricity [37]. All these abnormalities are potentially very costly to the plant breeding industry [38]. In our experimental procedure, as far as the propagation step is concerned, the best performance was achieved with no growth regulators added to the medium. Considering SEE, MS medium hormone free was the most effective as it facilitates high proliferation rates and shoot development.

Large-scale vegetative propagation of herbaceous and woody species depends on their rooting and acclimatization performances, which are influenced by several factors such as medium composition, mineral nutrition, light and temperature, and phytohormones [39,40]. In our investigation, particular attention was paid to the rooting performance of the new plantlets and, after a number of preliminary tests, a reliable protocol was developed. Initially, rooting of *C. maritima* was attempted using as auxin IAA supplemented at 5 and 10 μM. Rooting was obtained with both concentrations, but with significant differences, with 5 μM IAA being the best performing condition. This very promising result required a deeper investigation into its effectiveness, focusing on changes in usual cultural conditions. To evaluate if the culture support may positively influence the rooting performance, agar was replaced with zeolite. Usually, for micropropagation, agar is used as a support whereby nutrient media are solidified. Nevertheless, agar may cause some problems for in vitro plant development. Hyperhydricity, among others, is considered as a striking agar-related problem as it limits growth and multiplication in vitro, as well as establishment ex vitro [36,41]. To bypass this problem, some different inert materials have been used as a support, mostly for the acclimatization step [42–44]. The use of inert material such as sand or zeolite may be a low-cost agar substitute when the objective is the greatest plant production. Appreciable results were obtained with zeolite, a natural aluminium silicate with excellent ionic exchange properties and a high power of absorption. Percentages of both rooted and acclimatized plants were significantly higher than those obtained with agar as a support. This positive effect is probably due to the porous substrate that, as reported for other species too [44,45], significantly improves the root quality and, as a consequence, the number of acclimatized plants increases.

## 5. Conclusions

In this work, we present an experimental procedure to produce a high number of individuals of *C. maritima* independently from the natural life cycle of the species in the wild. The protocol reported can be successfully used to increase the production of new individuals for multipurpose applications. New plants were obtained in MS medium deprived of any plant growth regulator. The innovative application of zeolite during rooting significantly improves the percentage of plants usable for ex vitro establishment. The use of zeolite significantly reduces the costs of the entire process and improves the

effectiveness of the procedure, providing a valid alternative to the production of plants to be used in the pharmaceutical industry.

**Author Contributions:** Conceptualization, C.C., A.C. and G.G. methodology, C.C., A.C., A.M., L.A. and A.S.G. Investigation, C.C., A.C. and A.M. Data Curation, C.C. and A.C.; Writing—Original Draft Preparation, C.C. and A.C.; Writing—Review and Editing, L.A., F.C., A.S.G., S.P. and G.G.; Supervision, F.C. and G.G. All authors have read and agreed to the published version of the manuscript.

**Funding:** This work was funded by the Programme LIFE+, Project LIFE15 NAT/IT/000914 Cal.Mar.Si.— Measures of integrated conservation of *Calendula maritima* Guss., a rare and endangered species of the Sicilian vascular flora (https://lifecalmarsi.eu, accessed on 3 October 2022).

**Data Availability Statement:** Data are available from the corresponding author upon request.

**Conflicts of Interest:** The authors declare no conflict of interest.

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
