# Peer review of "Propagation of Calendula maritima Guss. (Asteraceae) through Biotechnological Techniques for Possible Usage in Phytotherapy"

_agronomy, doi:10.3390/agronomy12112788_

Round 1

Reviewer 1 Report

In general, this is an interesting study about biotechnological techniques of propagation of Calendula Maritima. The study is described clearly and adequately but some aspects of format should to be modified (detailed below).

- Tables should to be modified (i.e. readjusted the letter size and center it on the page).

- The title is so long. I do not suggest a change on it. However, it should be more sthetical avoiding the word partition.

- Overall, in the main text appears "P-value" indicating the "p" in upercase and in lowercase (ej.  lines 140, 141, 185 or 188). Please homogenise (I recommend the use of capitol letters. 

- Line 42. Please indicate "Calendula" in italics.

- Line 57. Please indicate "ex situ" in italics.

- Please indicate "in vitro" in italics in the complete manuscript (including figure captions).

Reviewer 2 Report

Dear Authors,

I think this paper represents a very good piece of research of both scientific and practical relevance. Nevertheless, even if I have no specific comments about the overall value of the research, perhaps it would be more appropriate to consider changing the final part of the title. In fact, it doesn’t seem to match with the declared aims of the paper. Thus, I would suggest this alternative title: Propagation of Calendula Maritima Guss. (Asteraceae) Through Biotechnological Techniques for safeguarding purposes, instead of Propagation of Calendula Maritima Guss. (Asteraceae) Through Biotechnological Techniques for Possible Usage in Phytotherapy. My note is about the circumstance that the word “Phytotherapy” only appears on the title, and nowhere else!

Please take also note of the following suggestions:

LINES:

40: I think it should be more correctly: C. suffruticosa subsp. fulgida

41: Consider to quote: Pasta, S., Garfì, G., Carimi, F., Marcenò, C. 2017 Rendiconti Lincei 28(2), pp. 415-424.

42: Calendula (italics)

50: Consider to quote: Pérez, M.P., Navas-Cortés, J.A., Pascual-Villalobos, M.J., Castillo, P. 2003 Plant Pathology 52(3), pp. 395-401.

55: in vitro (italics)

57: ex situ (italics)

57: only one full stop!

66&67: in vitro (italics)

80: Ronciglio, Maraone and Colombaia Sicilian Islets (consider also to include geographic coordinates)

81: Please specify the name of the person to whom the communication refers

106: in vitro (italics)

147: Table 1 (insert space)

164: I’d suggest: Significantly different roots are produced depending on the medium used

174-177: Scientific names and latin words in italics

Table 2: By HF do you mean Hormon Free? It was not previously specified in the text nor in the caption!

197: only one full stop!

211, 242, 244: italics

277: only one full stop! (Grammatico, F..)

280: insert comma after Raimondo

289: insert space before Applied

296: insert space before Carimi

273-377: In short, the entire References list must be thoroughly and carefully reviewed!

Reviewer 3 Report

Dear authors,

I have reviewed your manuscript Propagation of Calendula Maritima Guss. (Asteraceae) Through Biotechnological Techniques for Possible Usage in Phytotherapy" submitted for publication in Agronomy. The manuscript is quite interesting and presents a valuable collection of information on the optimization of a micropropagation procedure for Calendula maritima, a critically endangered plant species. An interesting strategy through use of mineral zeolite in the rooting phase was also presented.

I have listed a number of suggestions below:

In vitro should be standardized in italics throughout the manuscript.

Line 55: How about vegetative propagation using cuttings? You could also add a small comment on that.

Line 55: It would be great to have a sentence about in situ X ex situ conservation.

Line 56: Add a reference to support this sentence

Line 57: Add a reference to support this sentence

Line 67: In vitro multiplication was performed starting from nodal explants, but

Line 82: How big were the explants?

Line 84: Tween-20 (v/v)

Line 85: Did you rinse the explants with distilled water after ethanol treatment?

Line 87: Murashige and Skoog [11] solidified medium (MS)

Please use g L-1

Line 89: potassium hydroxide (KOH)

Line 89: Add PPM company, country. Did you add PPM after or before autoclaving? Add this information. Did you use test tubes for initiation or placed several explants in container?

Line 96: were excised from xx-week-old

Lines 97-98: “Explants were plated in MS medium supplemented with PPM 2% with or without 3µM phloroglucinol”. Consider rewriting the sentence, I can't understand it at this point.

Line 98: 3 µM

Line 102: shoot, and the

Line 113: How many shoots per Magenta™? Add this information

Line 114: Describe IAA. Add sucrose, pH, and agar

Line 121: 2% PPM, and 3 µM PG

Line 138: Please take a look at line 126 “formed roots were collected 15 days after rooting treatments”

Line 143: How about contamination? It would be great to have a sentence on that.

Line 148: This is the first time you mentioned MPG

Lines 148 and 149: (MS + 3 µ MPG)

Line 150: (3.4 cm)

Line 151: 3 µ MPG

Line 156: Please take a look at line 126 “formed roots were collected 15 days after rooting treatments”

Line 162: the value was about

Line 164: Significantly different roots are produced according to the support used; i.e., roots

Line 169: 15 cm

Line 169: The highest percentage.. add the value

Line 172: Delete “3.3. Figures, Tables”

Lines 174 and 177: C. maritima should be in italics. Add a scale to photos

Line 175: it is not possible to see the roots in the photo C; add a photo showing the roots.

Add the age of explants in each photo, i.e., In vitro rooting obtained on MS medium after x days/weeks of culture

Line 179: Root formation under different culture conditions after x days/weeks of culture.

Lines 180-181: 5 µM

Figure 2: Add a scale to photos

Table 1: MS + 3 µ MPG. Describe SEE and the treatments in the footnote of the table

Line 183: ….on axillary bud formation after x days/weeks of culture.

Line 187: ….on root formation x days/weeks of culture.

Line 190: Describe the treatments in the footnote of the table

Table 2: This is the first time you mentioned HF

Line 195: Add a reference

Line 197: …. several advantages [add references].

Lines 205, 207, and 210: Add a reference

Line 212: It would be great to have a sentence on ex situ conservation…

Line 218: It would be great to have a sentence about the importance of PPM. Add references on use of PPM on other crops. Consider adding: https://doi.org/10.3390/plants11192624 https://doi.org/10.1007/s00299-017-2185-1

Line 227: Add a reference

Line 233: by several factors such as xx, xx, and xx [references].

Line 237: concentrations at xx and xx.

Line 240: “The most significant change concerned the culture support”. Add more information on that, it is not clear.

Lines 254-259: Is it a conclusion? You can also describe the best procedure for micropropagation of this species here

Carefully check the formatting of the references following the Journal's guideline. There is a need to edit the references in the current version (MDPI | Reference List and Citations Style Guide)

 I support the publication of this manuscript after these minor revisions
